# Dance for people with chronic respiratory disease: a qualitative study

Keir Elmslie James Philip [iD] ,[1,2] Adam Lewis [iD] ,[3] Sian Williams,[1] Sara Catherine Buttery [iD] ,[1] Michael I Polkey,[1,2] William Man,[2] Daisy Fancourt [iD] ,[4] Nicholas S Hopkinson [iD] [1,2]

► Prepublication history and supplemental material for this paper are available online. To view these files, please visit the journal online (http://dx.doi.org/10.1136/bmjopen-2020-038719).

[1]National Heart and Lung Institute, Imperial College London, London, UK
[2]Respiratory Medicine, Royal Brompton & Harefield NHS Foundation Trust, London, UK
[3]Department of Health Sciences, Brunel University London, Uxbridge, UK
[4]Department of Behavioural Science and Health, University College London Research Department of Epidemiology and Public Health, London, UK

**Correspondence to**
Dr Keir Elmslie James Philip; k.philip@imperial.ac.uk

## ABSTRACT

**Objectives** To explore the experiences and perceived impact on health and well-being related to participation in a dance group for people with chronic respiratory disease (CRD).

**Design** An exploratory qualitative study using thematic analysis of semistructured interviews.

**Setting** A community dance group in a UK health centre.

**Participants** Convenience sample of long-term dance group participants.

**Intervention** Weekly community dance sessions designed for people with breathlessness, lasting 75 min, led by a trained community dance leader.

**Results** Convenience sample of eight participants, six females, aged 57–87 years (mean 75), with a median 2-year attendance at weekly dance sessions. Long-term attendance was driven by strongly held beliefs regarding the health and well-being benefits of participation. Four key themes were identified: dance as (1) a holistically beneficial activity, with physical and psychosocial health benefits including improved or maintained physical fitness and psychological well-being, and reduced need for healthcare; (2) an integral part of their life; (3) an enjoyable activity; and (4) a source of deep social cohesion.

**Conclusions** Dance group participants perceived a broad range of health benefits of relevance to the biopsychosocial impacts of their respiratory disease. The themes identified are useful in the ongoing planning and evaluation of dance as a holistic complex intervention for people with CRD. Further research is required to assess the extent of health impacts identified, and how dance might be most effectively placed as an option in the management of CRD.

**Trial registration number** NCT04006015.

## INTRODUCTION

Chronic respiratory diseases (CRD) are a leading cause of global morbidity and mortality,[1] often causing persistent and disabling symptoms including breathlessness, cough, skeletal muscle impairment and exercise limitation.[2 3] Additionally, psychosocial impacts such as depression, loneliness and social isolation are common[4 5] and are associated with worse clinical outcomes such as hospital admission for respiratory disease[6] and poorer physical performance.[7] Physical activity, exercise training and pulmonary rehabilitation (PR) are core components of CRD management.[8 9] However, engagement is inhibited by symptoms, availability, accessibility and attractiveness of local options.[10–12] Recently, interest in alternative forms of physical activity and exercise training has increased including tai chi,[13 14] yoga[15] and Singing for Lung Health (SLH).[16–20] Furthermore, the current focus on social prescribing means such approaches are of increasing interest.[21] Though an extensive body of research suggests art interventions can improve health and well-being,[22 23] respiratory specific arts-in-health research is currently limited. Nevertheless, evidence which exists suggests a range of biopsychosocial benefits.[24 25]

Dance has been shown to be beneficial in healthy people and individuals with a range of medical conditions. Impacts include improved physical performance, function, mood and social engagement, which are all highly relevant for people with CRD.[26–33] Furthermore, dance is a core aspect of societies and cultures globally, hence has great potential as an enjoyable, engaging intervention in respiratory care throughout the world.[34–36] Regarding dance for people with CRD, two feasibility studies found dance for CRD interventions to be feasible,[37 38] one also suggested dance may improve functional capacity, balance, anxiety and depression, physical activity and health-related quality of life.[38] However, no studies have assessed

participants' experiences and perceptions regarding health and well-being impacts, which is important given the need to understand the potential of dance for wider application and how to implement and evaluate it in trials and clinical practice.

### Strengths and limitations of this study

► The first study of long-term real-world participants of a community dance group for people with chronic respiratory disease and breathlessness.
► In-depth exploration of participant experience.
► The sample was from a single group which may limit transferability.
► Semistructured interviews facilitated a depth of understanding of clinically relevant topics, aiding the potential application of findings.

The objective of this study was to explore the experiences and perceived health and well-being impacts of participation in a community-based dance group for people with CRD—'Dance Easy'.

## Information on lead researcher

KEJP is a male, 33-year-old respiratory physician who previously worked as a dancer and dance teacher, with experience leading community dance and dance for people with respiratory disease. KEJP has received training in qualitative research methods from the Imperial College, University College London, and through self-directed learning. He is currently completing a PhD at Imperial College, using both qualitative and quantitative methods. Qualifications: MBChB, BSc, MRCP, EADTMH, DPMSA.

KEJP and AL are recreational dancers and previous dance experiences enabled discussions throughout the analysis to be explored fully and assumptions challenged within the researchers' assumptions of what dance is.

## METHODS
### Research design

We performed an exploratory qualitative study using thematic analysis within a critical realist paradigm, to explore the perceptions of participants of a community dance group for people with CRD. The primary focus was their views on the impact of participation on their health and well-being.

We used semistructured interviews (see topic guide in online supplemental appendix 1), developed by reviewing research on the impacts of conceptually related interventions such as PR, SLH and tai chi for people with CRD. This contributed deductive elements to the analysis. However, analysis was primarily inductive using open-ended questions and open coding. Participants were encouraged to continue discussing topics which appeared meaningful with prompts. Interviews were complemented by structured observations by KEJP, who participated by dancing with the group over four sessions. This participation aimed to build rapport and trust with participants to facilitate interviews, and make structured observations to contextualise interview content, and triangulate themes. Additionally, this method enabled greater reflexivity within the analysis. Participants were made aware that KEJP is a clinician (but not involved in their clinical care) with experience of community dance groups.

The eight regular group participants were approached face to face by the group leader, and all consented to participate (convenience sample) in the study. Eligibility criteria included a clinical diagnosis of CRD, age >18 years and able to consent. All participants received participant information sheets and provided signed informed consent.

## Dance group intervention

The group started in May 2017, with seven of the study participants present from the first few months. The group meets weekly in a community centre attached to a health centre in London. Participants usually meet in the café from 30 min before the session. The sessions are free for participants, the leader volunteers her time for free and the space is provided by Whittington Health. The original participants were attendees at a 'Breathe Easy' support group for people with respiratory conditions, additional members became aware of the dance group through word of mouth.

Sessions last approximately 75 min and include a warm-up, including qigong movements (see https://www.nqa.org/what-is-qigong- for more information on qigong), progressively complex group dance routines, warm-down, stretching and relaxation. The session leader (SW) is a trained community dance group leader with extensive experience working with people with CRD. Music is played throughout the sessions and includes an eclectic mix of genres, selected by the session leader and participants, to inspire dancing at an energy level to suit the group. Each track corresponds to a specific movement routine. Dances are selected from a repertoire, with new dances added gradually. Though led by SW, participants contribute their opinions, movements and artistic expressions.

The dances are sufficiently physically demanding to induce breathlessness, though not excessively, as judged by participants. Given the group's heterogeneity regarding disease severity, age and comorbidities, a degree of personalisation is required. This is achieved by providing optional variations on movements, and allowing participants to pace themselves and take breaks as required. An approach that sees the participant as the expert in their own condition and physical capacities, and trusts their ability to modify their exertion accordingly. Sessions are structured yet relaxed, focusing on dance as an enjoyable activity. Short breaks between dances are included for recovery and are used creatively to talk about dance, music and other topics that emerge. After the dance session, four to six participants go to a local pub to eat and chat, which developed spontaneously and became an established routine.

## Data collection

Convenience sampling was used. Interviews were conducted one to one by KEJP (see researcher information). Reflections and impressions were documented immediately following the interviews, with memo writing continuing throughout the analytical process. Interviews were conducted privately in the room where the dance takes place, or the café, depending on participant preference. Interviews, recorded by dictaphone, lasted 24–43 min, and consisted of open-ended questions, semistructured around the following topics:

- ► Experience of respiratory illness and impact on life.
- ► Experience of dance group.
- ► Perceived impacts of dance group participation.

## Data analysis

Interviews were transcribed verbatim by KEJP. KEJP and AL conducted thematic analysis based on the process outlined by Braun and Clarke[39] and Terry *et al*.[40] During phase 1, transcripts were read and re-read, and audio recordings listened to repeatedly. In phase 2, open inclusive coding of transcripts was used. This generated first cycle codes, mainly in vivo (verbatim), descriptive, process, emotive and value items. We chose this method to enable interpretive and more qualitative elements in our approach, which we felt was appropriate given the exploratory nature of this study. Second cycle coding using the context from the first cycle, then focused on discussions relating to dance, activity and their disease. Code mapping facilitated grouping of codes. Preliminary themes (phase 3) were constructed from code mapping and reorganisation in relation to the associated quotes. Phases 1–3 were completed by KEJP and AL independently, before coming together to alter, refine, consolidate and name themes (phase 4 and 5). Structured observations, reflexive accounts after interview and memos were incorporated to the thematic analysis. Themes were reviewed with the group leader SW. Participant respondent validation was not performed because of the value placed on researchers' interpretation within our methodology.

Demographic and disease-specific information was sought. Breathlessness scores were used to indicate severity, rather than lung function measures as breathlessness relates more closely to patient-related outcomes including mortality and functional impairment.[2] Furthermore, participants could provide this themselves, whereas hospital investigations were not readily accessible, and participants had different CRD, which limits the relevance of spirometry group averages to this study.

## Patient and public involvement

Patients and the public were not involved in the design of the study; however, the study specifically focuses on patient/participant perceptions.

## RESULTS

Eight regular dance group participants were recruited, representing the group at that time, including six women and two men, aged 57–87 years (mean 75). All reported CRD, primary diagnoses were reported as chronic obstructive pulmonary disease (COPD) (×5), asthma (×2), bronchiectasis (×1); however, multimorbidity was common including arthritis, diabetes, sarcoidosis, visual impairment and haemochromotosis. Four participants had a Medical Research Council breathlessness score of 2, one had a score of 3, three had a score of 1. None used supplementary oxygen. Participants lived in houses[6] and flats.[2] Half of the participants lived alone, the others cohabited with their partner or children. Interviews lasted a mean of 29 min 46 s (range 23 min 47 s to 43 min 14 s). Verbatim quotes are provided below to illustrate the themes. To facilitate understanding of verbatim terms that could be unfamiliar to non-native English speakers, explanations have been provided in square brackets.

No adverse events were reported. Participants perceived a wide range of health and well-being benefits related to attendance. All were extremely positive about the group, none expressed significant negatives, though some experienced transport-related challenges. Attendance was very good, both self-reported and according to the leader's register. Non-attendance was related to holidays, transport difficulties or health issues such as optician appointments. Participants showed marked determination regarding being physically active, including but not limited to dance (walking and gardening frequently mentioned), so that '*psychologically the illness has not taken over*' (participant 5). They perceived sedentarism as highly negative, often stating the only choice being to '*get on with it*' (participant 2). This determination and commitment was driven by the identified themes, which were dance as (1) a holistically beneficial activity; (2) an important part of their lives; (3) an enjoyable activity; and (4) a source of deep social cohesion. Themes 3 and 4 were drivers of themes 1 and 2, but also stood alone so were not considered subthemes (see figure 1).

## Dance as a holistically beneficial activity

Participants viewed the sessions as a combined exercise, dance and social activity. They described physical,

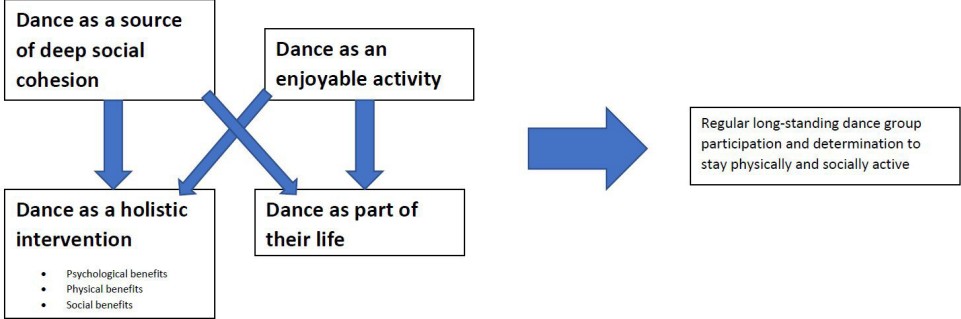

**Figure 1** A thematic map showing the main four themes and the key ways they relate to one another.

psychological and social benefits, which were synergistic and largely inseparable from one another.

> It's a bit of everything, afternoon social activity, because everyone has tea when they finish, or pub. It's a bit of exercise, a bit of, entertainment, it's a bit of, overall, learning. (Participant 6)

Participation was perceived as beneficial for overall health status—'*I feel like it's been a great benefit for me. It has impacted on me*' (participant 3). Impacts were described as improvements '*you're able to do things now*' (participant 3), or physical and cognitive maintenance in the context of ageing:

> …keeping one's body and one's mind active is very important. And that is really the be all of the end all [most important thing]. That's the object of the exercise. (Participant 2)

> It helps you maintain a standard as it were, of operation in the world I suppose. (Participant 2)

> And

> Also when the music comes on, you, about three beats and you're like, 'oh', so I like that because you're like, it's about maintaining that brain, that grey brain matter, yeah. And basically keeping you active. (Participant 6)

Perceived physical benefits included impacts on the disease and symptoms '*until I was doing the exercise it was bad*' (participant 7), and '*I think my breathing is better*' (participant 1). And implications for physical function '*mobility, I think, it's one of the things that I have found so much better. Because when I joined there were lots of things I found really difficult*' (participant 1).

Perceived mood and cognition benefits were also noted. For example, improvements in confidence,

> You feel you can do more things. So, if there's an offer coming up for an outing to a garden all day, I know that I'm going to go. (Participant 4)

> And

> Psychologically the illness has not taken over… because you're occupying your mind on something completely different, so you can't say 'oh I can't do that, I can't breathe', I can have a go. That's what life's all about at the end of the day. (Participant 5)

Impacts on mood were driven by enjoying participation (theme 3) and a sense of the group's deep social cohesion (theme 4), see figure 1 (thematic map). Additionally, participants 1, 3 and 5 perceived these holistic health benefits as leading to reduced healthcare utilisation, '*I haven't had a bad bronchitis episode since I've been doing it*' (participant 1), and '*you're not so prone to picking up all these chest infections*' (participant 3).

### Dance as an important part of their lives
Dance was perceived as an important part of their lives, '*It's just part of my life now*' (participant 3). For some

participants dance had been important throughout their lives (all but one reported dancing 'socially' at family events, three took classes), while for others, dance only gained importance since joining the group '*I was your typical wallflower …. I never really got into dancing at all. Till now*' (participant 3).

In general, the importance of dance participation was not related to their perceived ability or previous experience—'*I wouldn't call myself twinkle toes [a good dancer]*' (participant 3) and '*I can't dance for toffee [I can't dance well], and I can't sing, but hey (laughs) who cares*' (participant 1). However, perceptions regarding dance ability influenced how participants described what they do during the sessions. For example, some participants commented that they were not really dancing as they felt that a certain skill level was required for their individual definition of dance, instead describing the sessions as '*movement to music*' (participant 2). Nevertheless, learning to dance was frequently stated as a component of the sessions, hence differing definitions of dance likely represented differing life experiences of dance, rather than differing views on what they gained from participation in the group.

> I'm learning something new and it's also, you come in and learn a dance that six weeks ago you never thought you'd get your feet around [never thought you could learn the dance steps]. (Participant 6)

This satisfaction related to a sense of mastery was also important. Observation data also noted participants improved over the sessions, particularly completing routines with fewer errors, with a clear sense of achievement-related satisfaction, perhaps as a contrast to a background of decline in other aspects of their functioning. Structured observations also noted that participants were often quite elegantly dressed, which may not be expected in activities seen as purely exercise.

### Dance as an enjoyable activity
All participants greatly enjoyed the sessions, consistently describing it as fun, which was a main driver of attendance. This was repeatedly emphasised as being of major importance, and something participants felt was not appreciated or considered in other therapeutic exercise they had attended. In this sense, dancing was viewed as an end in itself, rather than the means to an end, as was the case for gym-based exercise training. Multiple factors contributed to their enjoyment including a light-hearted and non-judgemental atmosphere, in which participants felt active contributors rather than passive recipients of care.

> There is this immense feeling of contentment, and it's great. You know, when you've made an effort, you've done it, and jolly good. And that's it. (Participant 2)

The atmosphere in the sessions was very relaxed. The element of humour was highlighted in structured observations, with regular group laughter noted, with a very inclusive quality.

…before you know it, we're all in stiches [all laughing]. You know it's something, you know. Its promoted in the group. You can feel it. You get the buzz. You have a good laugh and, apart from doing your dancing you're having fun, even with the dancing you're having fun. (Participant 3)

Not feeling intimidated or judged was key to the enjoyment and contrasted with other exercise modalities and settings.

I won't go to the gym where there's a lot of younger people than myself. Because they can be intimidating. So, I won't do nothing like that. (Participant 5)

### Dance as a source of deep social cohesion

Participants described the other group members in very affectionate terms—'*I love everyone I'm with*', '*we're like a small knit family [close social relationship, care for each other like a family]*' (participant 6), and greatly valued their relationships with other individuals and the group as a whole.

It's not just about dancing it's about the whole, being part of the organisation, being part of a little unit that I think I enjoy coming out. (Participant 6)

Having shared experience was extremely important to participants. Participants commented on feeling 'other' when in public or with friends outside of the group. Paradoxically, although the group is built around having respiratory disease, not feeling 'other' created an environment where the disease could be forgotten, providing a form of respite.

I do it for the social side as well. Because it's very nice to meet people with the same basic difficulties. (Participant 2)

As a group, you know, we obviously bonded with each other, well we know each other, it is, you know it's something that…. I think we all enjoy it. (Participant 3)

We kind of understand each other when we can't breathe. (Participant 6)

But I think that for me it's about forgetting what's wrong with me until I actually go home. (Participant 6)

The depth of connection between participants underpinned the holistic impacts described in theme 1. The shared experience enabled participants to develop a combination of acceptance and increased motivation for improvement. This forged a collective self-motivation, with participants seeing their health as their own responsibility, directly related to the effort they put in '*you got to be up for it, you've got to be disciplined*' (participant 4). Yet, simultaneously appreciating that this could be best achieved together.

For me it's a good thing. It's basically being with likeminded people, people who understand how you

feel. You can see them, you can encourage them to do a little better. I think I've actually changed my attitude. Instead of being in denial about not being able to breathe, I think I've more embraced it. (Participant 6)

This deep social cohesion was attributed to the content of the sessions, particularly the presence of music, and dancing together.

Well it means you accept the whole thing. You're moving with a group of people who have a similar problem. (Participant 2)

The supportive atmosphere was also attributed to session leader (SW). SW is very supportive while challenging the participants to do the best they can, which is greatly appreciated by the participants who describe her as 'a lovely person'. There is a clear focus on abilities rather than impairment, which helps set the tone for the entire group.

…the wonderful thing is, nobody's ever been told 'you're useless and you can't do it'. (Participant 1)

(referring to the group leader's responses) It's the effort that one puts into it, and that is repaid enormously. (Participant 2)

The social relationships that have developed were also contrasted with other types of therapeutic exercise such as going to the gym, where they felt this depth of relationship was not created.

If you go to the gym, you're just doing your own thing, whereas here we are supporting each other. (Participant 1)

## DISCUSSION

This study found that long-standing participants in a dance group for people with CRD perceived a range of holistic health and well-being benefits. The dance group had become an important part of their lives. Enjoyment of participation and the development of deep social cohesion between participants were considered key drivers of these impacts.

This study has multiple strengths. To our knowledge it is the first study to explore community dance participation for individuals with respiratory disease in a real-world setting. A recent pilot study included qualitative aspects, based on informal group discussions and field notes,[37] but no other studies have used in-depth qualitative interviews with long-term attendees, outside of a trial setting, which improves the relevance of our work to real-world experience. Indeed, few qualitative studies of arts-in-health interventions in respiratory disease have explored such long-term participants' experiences. Second, the use of semistructured interviews enabled inductive exploration of specific topics of interest. Third, interpretation of the interview data was strengthened through use of structured observations.

Some limitations should be noted. First, sample size was limited by the number of group participants. However, given the exploratory nature of the study, sufficient data were collected to justify the themes. Second, as we only focused on a single group, the transferability of findings cannot be ascertained. Third, more detailed clinical characterisation of the participants' respiratory disease may have been of interest to people reading this study if considering practical implications for a specific context. However, we collected the relevant data for our research question. Fourth, a degree of selection bias may exist, as study participants were individuals who had maintained attendance, while those who did not enjoy the group no longer participated and were not included in the research. It would have been useful to interview people who no longer attended to explore the reasons for non-attendance. Similarly, the group has a small number of participants, which is likely due to no formal promotion activities because it is unfunded. Queries about the group are managed by the Breathe Easy team and no formal referral pathway yet exists. It is also possible the small group size may be small because potential participants did not see this as a desirable activity, but without further research it is not possible to draw any definite conclusions.

Our findings echo some of the findings from the limited existing research regarding dance for people with CRD. A recent study assessing a dance intervention for COPD showed it to be feasible, with secondary outcome measures suggesting improvements in functional capacity, balance, anxiety and depression, physical activity and health-related quality of life.[38] These findings suggest that elements of the perceived holistic benefits described in our study have quantifiable impacts using established assessment tools. Additionally, a second feasibility study assessing dance for people with breathlessness reported preliminary results suggesting dancing together is of importance, and participants reported 'coming alive'.[37] Such findings appear similar to themes 3 and 4 of our study. The themes identified here also map to multiple levels of Maslow's hierarchy of needs model of human motivation,[41] including safety, love and belonging, self-esteem and self-actualisation. This supports our interpretation that their experience as being complex and holistically beneficial drove their motivation for participation. Overall, our findings are broadly corroborative with existing studies, while deepening understanding of underlying drivers for participation and psychological benefits observed.

Research exploring the experiences of people with CRD participating in related activities shows some similarities and differences. Regarding PR, a high-value intervention[42] which commonly includes group-based callisthenic-type exercises, group support and increased self-confidence has been found to promote adherence.[43] However, lack of social support and overcoming the effort of living with COPD, as well as anxiety and depression[11] have been identified as negatively influencing participation in PR.[43] The participants in our study showed consistent attendance over an extended period, despite often living alone or reporting limited social contacts. Potential explanations include the dance group providing an experience sufficiently different from PR to overcome these issues, with participants reporting strong social connectiveness with peers, and high levels of confidence in their abilities, despite the evident disease-based limitations. A related activity is SLH which consists of specially developed group singing sessions for people with CRD. Research on SLH has found very similar impacts to the current study, including positive impacts on general well-being, community/social support and enjoyment of participation. Both suggest physical benefits.[17–19] Our findings echo those of research on community dance interventions in different patient groups including improving physical performance, mood, cognition, quality of life,[31 44 45] sense of joy and pride,[46] interpersonal relationships and connections.[47] Hence, the perceived health impacts here reported may not be disease specific, but rather represent the impact of group dance participation on people in general. That said, given the biopsychosocial impacts of CRD and relative lack of effective management options for psychosocial components, dance interventions appear particularly well suited for people with CRDs.

Of note, the study participants did not report severe breathlessness, none used supplementary oxygen and all lived relatively independently. Given the strong evidence base for PR and other forms of exercise in people with more severe breathlessness and disease-related impairment, it would be of interest to explore the use of dance interventions in such individuals. Similarly, they were generally quite active people, who clearly valued various types of physical activity, this may frame their highly positive perspectives, and indicates further research on interventions targeting less active individuals would be of value, as well as groups who meet via the internet.

## CONCLUSION

CRDs are common and result in a high burden of morbidity, mortality and related economic implications. Therefore, low-cost, low-resource interventions with holistic impacts on health and well-being should be of broad interest. In particular, the perceived impacts on physical performance, psychosocial health and health resource utilisation are promising. Furthermore, this study highlights the importance of developing enjoyable physical activities that provide opportunity for self-expression for people with CRD to use intrinsic motivation, as an end in itself, but also to support compliance.

Participants perceived a broad range of positive holistic health and well-being benefits; viewed participation as an important part of their lives; greatly enjoyed participation; and felt deep social cohesion with other participants. Further research is required to confirm, clarify and quantify these impacts, and assess the cost-effectiveness of sustainable methods of delivery including funding for session leaders and an appropriate dance space.

**Acknowledgements** We would like to give our thanks to the study participants for their time and effort.

**Contributors** KEJP had the original idea for the study, led the study including design and ethical approval, and conducted and transcribed the interviews. KEJP and AL conducted the analysis and wrote the first draft of the manuscript. SW helped develop the study concept and focus, the content of the topic guide, and reviewed and suggested improvements to manuscript drafts. SCB, MIP, WM, DF and NSH provided valuable input in developing the methodological approach, developing the protocol and gaining ethical approval. All authors (KEJP, AL, SW, SCB, MIP, WM, DF, NSH) contributed to the study design, writing, reviewing and editing the manuscript, and approved the final manuscript for submission.

**Funding** KEJP was supported by the Imperial College Clinician Investigator Scholarship. DF was supported by the Wellcome Trust (205407/Z/16/Z).

**Disclaimer** The funders had no say in the design and conduct of the study; collection, management, analysis and interpretation of the data; preparation, review or approval of the manuscript; and decision to submit the manuscript for publication.

**Competing interests** SW is the founder of the dance group and leads the sessions. She was not involved in the thematic analysis.

**Patient consent for publication** Not required.

**Ethics approval** Ethical approval was granted by the Office for Research Ethics Committees Northern Ireland (ORECNI) (19/NI/0073).

**Provenance and peer review** Not commissioned; externally peer reviewed.

**Data availability statement** No data are available. Data from this study are not being made available for sharing as it would not be possible to anonymise the interview transcripts given the personal content included and the very limited number of dance groups for respiratory conditions in London.

**ORCID iDs**
Keir Elmslie James Philip http://orcid.org/0000-0001-9614-3580
Adam Lewis http://orcid.org/0000-0002-0576-8823
Sara Catherine Buttery http://orcid.org/0000-0001-9410-414X
Daisy Fancourt http://orcid.org/0000-0002-6952-334X
Nicholas S Hopkinson http://orcid.org/0000-0003-3235-0454

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
