## [Reviewer comments · BMJ Open]

ARTICLE DETAILS

TITLE (PROVISIONAL)	Dance for people with chronic respiratory disease: A qualitative study
AUTHORS	Philip, Keir; Lewis, Adam; Williams, Sian; Buttery, Sara; Polkey, Michael; Man, William; Fancourt, Daisy; Hopkinson, Nicholas

VERSION 1 – REVIEW

REVIEWER	Rainer Gloeckl Schoen Klinik Berchtesgadener Land, Schoenau am Koenigssee, Germany
REVIEW RETURNED	19-Apr-2020

GENERAL COMMENTS	Philip and colleagues performed a qualitative study on the experiences of dancing in patients with chronic respiratory diseases. They analyzed semi-structured interviews of eight participants from a dance group specified for patients with chronic respiratory diseases. The semi-structured interviews and its analysis were performed with an excellent methodological standard. Although it was only a small sample and only participants from a single dance group were interviewed the authors summarized a comprehensive insight into the patients' experiences identifying four key themes. I have only a few minor comments: - It seems that all participants of the dance group had a mild CRD (mMRC mostly 1 to 2, no patient needed supplemental oxygen). Could this kind of CRD-specific dance group also be feasible for patients with moderate to severe CRDs? Please add some thoughts and experiences in the discussion section.- The patients' experiences were compared with the experiences of singing groups. It was concluded that they were similar. Please add also qualitative studies on patients experiences for exercise or calisthenics groups in the discussion section- How physically active were the participants beyond dancing? Did they engage in any further exercise activities?- Is the dance group limited to 8 people or why did no further patients join the group within the last 2 years?- It was mentioned that some participants had former dance experiences. Please add the numbers for "former" and "new" dancers. How did the "new" dancers got aware of the dance group? Did their motivation differ from the ones that already liked to dance before?- One of the key themes is that the dance group got "an important part of their lives". It would be ideal if we could identify individual physical activities for each CRD patient that could become an important part also in their life. Identifying and providing physical activities that an individual patient likes (high intrinsic motivation)
--

	would probably increase the long-term adherence to this activity. Could we transfer something from this study for the general motivation in CRD patients? - Figure 1 is a bit confusing on the first look. Maybe it can be structured differently. I would suggest do delete the theme numbers in the figure.
--	---

REVIEWER	Hilary Bungay Anglia Ruskin University United Kingdom I know one of the authors Daisy Fancourt but have not discussed this study with her or had any prior knowledge of this research.
REVIEW RETURNED	06-May-2020

GENERAL COMMENTS	Comments to the author Thank you for submitting this paper for review, it is an interesting topic area and I would like to see the study published following some revisions. There is no mention of any ethical review process in the paper or statements about ethical review in the documentation I received but I would assume as a PhD student that this research would have been through some university ethical review process? Abstract I suggest that you frame this as an Exploratory qualitative design as this would to some extent remove the need to justify your sample size using the sentence on page 10 line 48. "However it was decided that sufficient data were collected to justify the themes" as you need to have some justification for taking that decision if you are going to retain that statement. Under the Conclusions: line 32 recommend removal of 'and other arts interventions' as it doesn't seem to follow from previous statement? Main Text As above, I think this is an exploratory qualitative study. Page 5 line 6 remove 'in' Under Dance group intervention 'but not feel too fast' (line 21) 'though not excessively' (line 25) are very subjective terms – it would be helpful to have some clarification as to what is meant. Line 34 'became established' this sentence need to be completed for example an established social event/outing or ritual? Data analysis Line 53: 'Interviews were transcribed verbatim by KP' is a repeat from line 52. Although it is stated that Braun and Clarke's thematic analysis approach is used the language describing the analytical process is not consistent with their work. For example, Braun and Clarke do not refer to 'code clustering' or 'meaningful associated units', and open coding and in-vivo are more common in reporting grounded theory data analysis. In addition, in this section the statement on p6 line 8-9 referring to participant respondent validation... etc. suggests a more interpretivist approach used in the study design. Page 5 line 15 change 'interpretation' to 'relevance' Page 7 lines 21-23 here it refers to themes 3 and 4 being 'drivers' of themes 1 and 2 _ I think the thematic map (Figure 1) should indicate this in some way to show that they are linked. Also experiencing deep social cohesion would have a psychological benefit too – in the sense of belonging and so those should also be linked on the map.
---

	Page 8 line 42 this should read 'fewer' not 'less' errors Page 8 line 46 – not purely exercise but also a social event? Discussion The limitations section needs revision Line 48: as above on what basis was this taken? –Reframe this as an exploratory study as you interviewed all the participants in the group Line 50 the third point who would have been more interested in the clinical characterisation of the disease – does this matter for this project? Line 51: “However we feel this was adequate ...” – revise – you collected the data that was relevant to your research question. Line 52: This fourth point is important, the fact they were long term attenders would indicate they value the sessions and would report positive impacts. It would have been useful to interview those who did not continue coming to find out why- I am not suggesting this should be done for this study but this should be included in response to this limitation. Conclusion There is an issue with sustainability of such groups – the dance teacher does this for free and there is not costs for room hire. This would not be the case in many areas, future research could also look at sustainability and costs of such interventions.
--	---

REVIEWER	Jana De Brandt Hasselt University Belgium
REVIEW RETURNED	19-Jun-2020

GENERAL COMMENTS	Dear authors, I have read the article with great interest and joy. I am very happy to understand that dancing can give a group of patients with CRD the tools to stay physically active and mentally well. The manuscript is overall very well and clearly written. Therefore I advise a minor revision purely based on some small comments below. Good luck! Introduction: - Majority of references of the first paragraph is based on research within COPD. As you include patients with a variety of CRD, it would be preferable that you also include some references including other respiratory pathologies than COPD. Methods: - Would it be possible to explain or give examples of what qigong movements are? Maybe pictures in supplemental material? As I realize that a majority of readers will not know what qigong movements are. - page 5 - line 53: repetition of the first sentence of the paragraph 'Interviews were transcribed verbatim by KP, then KP and AL conducted further analysis.' Please remove the sentence. Results: - Table 1: Interview (min/s) = is the unit minute per session? If so, please write full, as this is not clear.
---

	 - Table 1: patient 1: in column house/flat cohabitating?: here you write a ',' after house, while in patient 2, you write a ' '. - Table 1: please provide a more extensive caption for table 1. Please explain abbreviations used in the table (M, F, N, MRC, A1ATD). - Table 1: column diagnosis + comorbidities: sometimes you start a new diagnosis or comorbidity with a capital letter and sometimes you don't. Please be consistent. - Table 1: what is a maisonette? - Figure 1: it is correct that there are no key ways (arrows) directed to theme 2? - Page 9: line 42 - 43: 'And that is really the be all of the end all.' - Is it possible to make this sentence more understandable for the reader? As non-native English speaker I don't really understand what the participant wants to say. - In general I really like the fact that you take over the exact sayings of the participants. But for non-native English speakers/readers some of the expressions used by the participant might be difficult to understand. Is there any solution for this? - page 11 - line 31: do you mean 'being different' with the word 'other'? If so, I would suggest to change this word.
--	--

VERSION 1 – AUTHOR RESPONSE

Reviewer(s)' Comments to Author:

Reviewer: 1

Reviewer Name: Rainer Gloeckl

Institution and Country: Schoen Klinik Berchtesgadener Land, Schoenau am Koenigssee, Germany

Please state any competing interests or state 'None declared': None declared

Please leave your comments for the authors below Philip and colleagues performed a qualitative study on the experiences of dancing in patients with chronic respiratory diseases. They analyzed semi-structured interviews of eight participants from a dance group specified for patients with chronic respiratory diseases.

The semi-structured interviews and its analysis were performed with an excellent methodological standard. Although it was only a small sample and only participants from a single dance group were interviewed the authors summarized a comprehensive insight into the patients' experiences identifying four key themes.

I have only a few minor comments:

- It seems that all participants of the dance group had a mild CRD (mMRC mostly 1 to 2, no patient needed supplemental oxygen). Could this kind of CRD-specific dance group also be feasible for patients with moderate to severe CRDs? Please add some thoughts and experiences in the discussion section.

Added a comment to the end of the discussion suggesting this could be explored in further research.

- The patients' experiences were compared with the experiences of singing groups. It was concluded that they were similar. Please add also qualitative studies on patients experiences for exercise or calisthenics groups in the discussion section

Many thanks for this comment. Relevant qualitative studies regarding pulmonary rehabilitation, had been included, however, we have changed the wording to emphasis the calisthenic/exercise components of PR as the reason for referencing these papers and making these comparisons with the dance activity.

- How physically active were the participants beyond dancing? Did they engage in any further exercise activities?

The participants were quite active people and mentioned participation in other types of physical activity (especially walking and gardening). We have added to the second paragraph of the results 'Participants showed marked determination regarding being physically active, including but not limited to dance (walking and gardening frequently mentioned),....'

We have also mentioned this point in the discussion as a potential consideration/limitation, stating 'they were generally quite active people, who cleared valued various types of physical activity, this may frame their highly positive perspectives, and indicates further research on interventions targeting less-active individuals would be of value.'

- Is the dance group limited to 8 people or why did no further patients join the group within the last 2 years?

The group is not formally limited in numbers, though due to nature of the dance group and available space, substantial increases in participant numbers would not be feasible in the current setting. Additionally, as the group has no specific funding, no formal publicity has taken place. There have been a small number of people who tried the sessions, but did not keep up attendance. To acknowledge these considerations, in the limitations it states 'a degree of selection bias may exist, as study participants were individuals who had maintained attendance, while those that did not enjoy the group no longer participated and were not included in the research', and we have added 'Similarly, the group has a small number of participants, which is likely due to no formal promotion activities because it is unfunded and the queries are managed by the Breathe Easy team themselves. No formal referral pathway yet exists. It is possible the small group size may suggest that some potential participants did not see this as a desirable activity. However, given the lack of promotion it is not possible to say.'

- It was mentioned that some participants had former dance experiences. Please add the numbers for "former" and "new" dancers. How did the "new" dancers got aware of the dance group? Did their motivation differ from the ones that already liked to dance before?

Categorising the participants as former or new dancers is difficult, as most are of a generation where social dancing was the norm. To clarify this point we have added '(all but one reported dancing

'socially' at family events, 3 took classes)' to the results under 'Dance as an important part of their lives'.

We have added the following paragraph to clarify that previous experience did not impact enjoyment/motivation with the sentence 'the importance of dance participation was not related to their perceived ability or previous experience'.

Also, in the methods section, that 'The original participants were attendees at a 'Breathe Easy' support group for people with respiratory conditions, additional members became aware of the dance group through word of mouth.'

- One of the key themes is that the dance group got "an important part of their lives". It would be ideal if we could identify individual physical activities for each CRD patient that could become an important part also in their life. Identifying and providing physical activities that an individual patient likes (high intrinsic motivation) would probably increase the long-term adherence to this activity. Could we transfer something from this study for the general motivation in CRD patients?

The intended meaning of this comment it is not completely clear to us. The dance group itself was the individual physical activity that had become important in the participants lives. And, it is that enjoyment (fun and social) that we have proposed as driving the long-term adherence in the dance group, due to the intrinsic motivation. This is supported by the first second paragraph of the results section.

Regarding transferring something from this study for the general motivation in CRD patients, we feel (as you suggest) that this study highlights the importance of developing enjoyable physical activities for people with CRD to utilise intrinsic motivation, as an end in itself, but also to support compliance. As such we have added a statement to that effect to the conclusion.

Apologies if we have misinterpreted the comment.

- Figure 1 is a bit confusing on the first look. Maybe it can be structured differently. I would suggest do delete the theme numbers in the figure.

Many thanks, we have updated the figure to make it clearer.

Reviewer: 2

Reviewer Name: Hilary Bungay

Institution and Country:

Anglia Ruskin University

United Kingdom

Please state any competing interests or state 'None declared': I know one of the authors Daisy Fancourt but have not discussed this study with her or had any prior knowledge of this research.

Please leave your comments for the authors below Comments to the author

Thank you for submitting this paper for review, it is an interesting topic area and I would like to see the study published following some revisions.

There is no mention of any ethical review process in the paper or statements about ethical review in the documentation I received but I would assume as a PhD student that this research would have been through some university ethical review process?

Thank you for highlighting this. The following statement has been added 'Ethical approval was granted by the Office for Research Ethics Committees Northern Ireland (ORECNI) (19/NI/0073).'

Abstract

I suggest that you frame this as an Exploratory qualitative design as this would to some extent remove the need to justify your sample size using the sentence on page 10 line 48. "However it was decided that sufficient data were collected to justify the themes" as you need to have some justification for taking that decision if you are going to retain that statement.

We agree that this describes the type of study more accurately. As such, we have updated the methods section (abstract and main text) to describe this as an exploratory qualitative study, and edited the comment you raised to the following 'However, given the exploratory nature of the study, it was decided that sufficient data were collected to justify the themes.'

Under the Conclusions: line 32 recommend removal of 'and other arts interventions' as it doesn't seem to follow from previous statement?

Agreed and removed.

Main Text

As above, I think this is an exploratory qualitative study.

Changed as stated above.

Page 5 line 6 remove 'in'

Removed.

Under Dance group intervention 'but not feel too fast' (line 21) 'though not excessively' (line 25) are very subjective terms – it would be helpful to have some clarification as to what is meant.

We have added clarification, that these are deliberately subjective terms, relating to the participants' experience, which is used to titrate the intensity. To further clarify the reasoning we have added the following 'An approach that sees the participant as the expert in their own condition and physical capacities, and ability to titrate their exertion accordingly.'

Line 34 'became established' this sentence need to be completed for example an established social event/outing or ritual?

Changed to '...became an established routine.'

Data analysis

Line 53: 'Interviews were transcribed verbatim by KP' is a repeat from line 52.

Removed.

Although it is stated that Braun and Clarke's thematic analysis approach is used the language describing the analytical process is not consistent with their work. For example, Braun and Clarke do not refer to 'code clustering' or 'meaningful associated units', and open coding and in-vivo are more common in reporting grounded theory data analysis.

Thank you for raising this as a potential point of confusion. We have removed the terms 'clustering' and 'meaningful associated units' as Braun and Clarke don't use these terms, and have reworded as 'grouping of codes' which is a more accurate description of our approach.

Additionally, we have added a further reference to work by Terry et al(1) which explains that, although inductive approaches are more common in grounded theory, both inductive and more theoretical approaches can be applied in thematic analysis. . In relation to these inductive elements of the analysis, we state that 'open inclusive coding' was used which included in-vivo codes (amongst others such as descriptive, emotive and value codes), which were descriptions of the coding types which became apparent as a result of the analysis. These resulted from not restricting the types of codes we were going to use prior to starting the analysis, as we felt for this project, utilising the flexible approach to coding and theme development described by Terry et al. was particularly appropriate. We provide details of the researchers involved in the analysis in order to provide transparently to the reader, their background and interest in dance which was important in reflexive dialogue.

In addition, in this section the statement on p6 line 8-9 referring to participant respondent validation... etc. suggests a more interpretivist approach used in the study design.

Though member checking can be used in Thematic Analysis or more interpretive approaches, we decided that this was not required in this case. We included a statement regarding it not being included as this is part of the COREQ checklist regarding the reporting of qualitative research, so that people reading the manuscript could see that we had made a considered decision regarding this point.

Page 5 line 15 change 'interpretation' to 'relevance'

Good suggestion, we have made this change.

Page 7 lines 21-23 here it refers to themes 3 and 4 being 'drivers' of themes 1 and 2 _ I think the thematic map (Figure 1) should indicate this in some way to show that they are linked. Also experiencing deep social cohesion would have a psychological benefit too – in the sense of belonging and so those should also be linked on the map.

We have revised the thematic map, which now illustrates these relationships more clearly.

Page 8 line 42 this should read 'fewer' not 'less' errors Page 8 line 46 – not purely exercise but also a social event?

Changed.

Discussion

The limitations section needs revision

Line 48: as above on what basis was this taken? –Reframe this as an exploratory study as you interviewed all the participants in the group Line 50 the third point who would have been more interested in the clinical characterisation of the disease – does this matter for this project?

As suggested, we have reframed as an exploratory study.

Though more detailed clinical characterisation of disease did not matter for this project, it may be of interest and relevance for people reading the study when considering practical implications in their specific context. For this reason we have included this as a potential limitation.

Line 51: "However we feel this was adequate ..." – revise – you collected the data that was relevant to your research question.

We have clarified this as suggested.

Line 52: This fourth point is important, the fact they were long term attenders would indicate they value the sessions and would report positive impacts. It would have been useful to interview those who did not continue coming to find out why- I am not suggesting this should be done for this study but this should be included in response to this limitation.

We have included and further expanded this point with the addition of the following to this section 'It would have been useful to interview people who no longer attended to explore the reasons for non-attendance. Similarly, the group has a small number of participants, which is likely due to no formal promotion activities, referral pathways, or relationships with healthcare providers. However, the small group size may suggest that other potential participants did not see this as a desirable activity.'

Conclusion

There is an issue with sustainability of such groups – the dance teacher does this for free and there is not costs for room hire. This would not be the case in many areas, future research could also look at sustainability and costs of such interventions.

Thank you we have added the following to the conclusion 'and assess the cost effectiveness of sustainable methods of delivery (i.e. paid session leaders and funding for space).

Reviewer: 3

Reviewer Name: Jana De Brandt

Institution and Country:

Hasselt University

Belgium

Please state any competing interests or state 'None declared': None declared

Please leave your comments for the authors below

Dear authors,

I have read the article with great interest and joy. I am very happy to understand that dancing can give a group of patients with CRD the tools to stay physically active and mentally well. The manuscript is overall very well and clearly written. Therefore I advise a minor revision purely based on some small comments below. Good luck!

Introduction:

- Majority of references of the first paragraph is based on research within COPD. As you include patients with a variety of CRD, it would be preferable that you also include some references including other respiratory pathologies than COPD.

Many thanks for raising this. We have added in some references to other specific CRD, and selected broader references for topics such as pulmonary rehabilitation which discuss CRD rather than COPD alone. Many of references remain predominantly COPD focused, as COPD is the primary CRD diagnosis in our participants, we feel this new balance of references is appropriate.

Methods:

- Would it be possible to explain or give examples of what qigong movements are? Maybe pictures in supplemental material? As I realize that a majority of readers will not know what qigong movements are.

Good idea. We have opted for adding a '(see <https://www.nga.org/what-is-qigong/> for more information on qigong), as the manuscript is all ready quite long, so we are hesitant about overloading readers.

- page 5 - line 53: repetition of the first sentence of the paragraph 'Interviews were transcribed verbatim by KP, then KP and AL conducted further

analysis.' Please remove the sentence.

Removed.

Results:

- Table 1: Interview (min/s) = is the unit minute per session? If so, please write full, as this is not clear.

We have opted to remove table 1 and put summarised group averages on advice of the journal editorial team.

- Table 1: patient 1: in column house/flat cohabitating?: here you write a ',' after house, while in patient 2, you write a '.'

We have opted to remove table 1 and put summarised group averages on advice of the journal editorial team.

- Table 1: please provide a more extensive caption for table 1. Please explain abbreviations used in the table (M, F, N, MRC, A1ATD).

We have opted to remove table 1 and put summarised group averages on advice of the journal editorial team.

We have explained 'MRC' where used in the text.

- Table 1: column diagnosis + comorbidities: sometimes you start a new diagnosis or comorbidity with a capital letter and sometimes you don't. Please be consistent.

We have opted to remove table 1 and put summarised group averages on advice of the journal editorial team.

- Table 1: what is a maisonette?

We have opted to remove table 1 and put summarised group averages on advice of the journal editorial team.

- Figure 1: it is correct that there are no key ways (arrows) directed to theme 2?

We have revised and simplified the figure which is now clearer.

- Page 9: line 42 - 43: 'And that is really the be all of the end all.' - Is it possible to make this sentence more understandable for the reader? As non-native English speaker I don't really understand what the participant wants to say.

I can see how this would be difficult to understand, particularly as the participant did not use the usual wording for the phrase. We have added '[most important thing]' after the phrase to explain its meaning. We have not changed the participants words.

- In general I really like the fact that you take over the exact sayings of the participants. But for non-native English speakers/readers some of the expressions used by the participant might be difficult to understand. Is there any solution for this?

We are glad that you like the verbatim use of quotes. We feel this is important to ensure the participant's voice is heard. We would be happy to put further clarifications regarding specific statements (as above), but its difficult for us to know exactly what needs further clarification without highlighting specific phrases.

- page 11 - line 31: do you mean 'being different' with the word 'other'? If so, I would suggest to change this word.

We have changed this as suggested to add clarity.

References:

1. Terry G, Hayfield, N., Clarke, V., Braun, V.,. Thematic Analysis in: The SAGE Handbook of Qualitative Research in Psychology. London: SAGE Publications Ltd; 2017.

VERSION 2 – REVIEW

REVIEWER	Rainer Gloeckl Schoen Klinik Berchtesgadener Land, Schoenau, Germany
REVIEW RETURNED	20-Jul-2020

GENERAL COMMENTS	I thank the authors for their comprehensive revision and advice to accept the manuscript for publication.
---

REVIEWER	Hilary Bungay Anglia Ruskin University England
REVIEW RETURNED	17-Jul-2020

GENERAL COMMENTS	Thank you for revising and re-submitting this manuscript for review. The issues raised by the reviewers appear to have been mostly addressed but there remains a few minor points and some minor typographic errors have crept into the revised manuscript. Page 5 line 29/30 to address point raised previously regarding the subjective nature of 'feel too fast' the revised wording in brackets '(subjectively from the participant's perspective)' doesn't really clarify the issue as each participant may have a different perspective as to what is 'too fast'. Could this sentence be revised – for example 'selected by the session leader and participants, to inspire dancing at an energy level to suit the group'?
---

	Page 5 line 41 – titrate is a very clinical term – suggest ‘modify their exertion’ would read better Page 6 under Data Analysis lines 6-8 read very clumsily. Suggest instead.... “Interviews were transcribed verbatim by KP. KP and AL conducted thematic analysis based on the process outlined by Braun and Clarke (36) and Terry et al. (37). During phase 1....” This also saves some words. Page 6 line 13 “Second cycle coding using...” rather than ‘used’. Page 6 line 24 should ‘Breathlessness was to indicate severity’ read ‘Breathlessness scores were used to indicate severity’? Page 7 the list of themes should be linked to thematic map e.g. on line 58 after ‘considered subthemes’ write (see Figure 1), as currently the thematic map is not referred to until line 50 on page 8. Page 11 line 24 – delete ‘it was decided that’ Page 11 line 29 rather than ‘their context’ suggest ‘a specific context’ Page 11: lines 35-39 suggest rewording to “Similarly, the group has a small number of participants, which is likely due to there being no formal promotional activities because it is unfunded. Queries about the group are managed by the Breathe Easy team and no formal referral pathway yet exists. It is also possible the group size maybe small because potential participants did not see this as a desirable activity, but without further research it is not possible to draw any definite conclusions.” Page 11 line 48 ‘suggesting dancing’ rather than ‘suggestive dancing’ Page 12 last line of conclusion remove brackets (i.e. paid session leaders and funding for space) and instead... ‘sustainable methods of delivery including funding for session leaders and an appropriate dance space’.
--	--

REVIEWER	Jana De Brandt UHasselt, Belgium
REVIEW RETURNED	28-Jul-2020

GENERAL COMMENTS	Dear authors, Thank you for adapting the manuscript based on the reviewers suggestions. For me the manuscript is ready for publication after adapting the following comment. Thank you and good luck. Regarding one of my comments: Comment: In general I really like the fact that you take over the exact sayings of the participants. But for nonnative English speakers/readers some of the expressions used by the participant might be difficult to understand. Is there any solution for this? Response: We are glad that you like the verbatim use of quotes. We feel this is important to ensure the participant’s voice is heard. We would be happy to put further clarifications regarding specific statements (as above), but its difficult for us to know exactly what needs further clarification without highlighting specific phrases. Comment: I would address the following sentences in the same way as you did with the previous sentence (using [...] for an
---

	explanation) to ensure all non-native English understand the sentences. I have put the more difficult wordings (in my opinion) after the arrow.  - 'I wouldn't call myself twinkle toes' (participant 3) ==> 'twinkle toes' - 'I can't dance for toffee, and I can't sing, but hey (laughs) who cares.' (participant 1). ==> 'can't dance for toffee' - 'I'm learning something new and it's also, you come in and learn a dance that six weeks ago you never thought you'd get your feet around.' ==> 'you'd get your feet around' - '...before you know it, we're all in stiches. You know it's something, you know. Its promoted in the group. You can feel it. You get the buzz. You have a good laugh and, apart from doing your dancing you're having fun, even with the dancing you're having fun.' (participant 3) ==> 'we're all in stiches" - 'we're like a small knit family' ==> 'knit family'
--	---

VERSION 2 – AUTHOR RESPONSE

Reviewer(s)' Comments to Author:

Reviewer: 2

Reviewer Name

Hilary Bungay

Institution and Country

Anglia Ruskin University

England

Please state any competing interests or state 'None declared':

None declared

Please leave your comments for the authors below Thank you for revising and re-submitting this manuscript for review.

The issues raised by the reviewers appear to have been mostly addressed but there remains a few minor points and some minor typographic errors have crept into the revised manuscript.

Page 5 line 29/30 to address point raised previously regarding the subjective nature of 'feel too fast' the revised wording in brackets '(subjectively from the participant's perspective)' doesn't really clarify the issue as each participant may have a different perspective as to what is 'too fast'. Could this sentence be revised – for example 'selected by the session leader and participants, to inspire dancing at an energy level to suit the group'?

- *Thank you for this suggestion, we have reworded as suggested.*

Page 5 line 41 – titrate is a very clinical term – suggest 'modify their exertion' would read better

- *Thank you for this suggestion, we have reworded as suggested.*

Page 6 under Data Analysis lines 6-8 read very clumsily. Suggest instead...

"Interviews were transcribed verbatim by KP. KP and AL conducted thematic analysis based on the

process outlined by Braun and Clarke (36) and Terry et al. (37). During phase 1....”
This also saves some words.

- *Thank you for this suggestion, we have reworded as suggested.*

Page 6 line 13 “Second cycle coding using...” rather than ‘used’.

- *Thank you for this suggestion, we have reworded as suggested.*

Page 6 line 24 should ‘Breathlessness was to indicate severity’ read ‘Breathlessness scores were used to indicate severity’?

- *Thank you for this suggestion, we have reworded as suggested.*

Page 7 the list of themes should be linked to thematic map e.g. on line 58 after ‘considered subthemes’ write (see Figure 1), as currently the thematic map is not referred to until line 50 on page 8.

- *Thank you for this suggestion, we have added a reference to Figure 1 to the end of the second results paragraph.*

Page 11 line 24 – delete ‘it was decided that’

- *Removed as suggested.*

Page 11 line 29 rather than ‘their context’ suggest ‘a specific context’

- *Thank you for this suggestion, we have reworded as suggested.*

Page 11: lines 35-39 suggest rewording to “Similarly, the group has a small number of participants, which is likely due to there being no formal promotional activities because it is unfunded. Queries about the group are managed by the Breathe Easy team and no formal referral pathway yet exists. It is also possible the group size maybe small because potential participants did not see this as a desirable activity, but without further research it is not possible to draw any definite conclusions.”

- *Thank you for this suggestion, we have reworded as suggested.*

Page 11 line 48 ‘suggesting dancing’ rather than ‘suggestive dancing’

- *Edited.*

Page 12 last line of conclusion remove brackets (i.e. paid session leaders and funding for space) and instead... ‘sustainable methods of delivery including funding for session leaders and an appropriate dance space’.

- *Thank you for this suggestion, we have reworded as suggested.*

Reviewer: 1

Reviewer Name
Rainer Gloeckl

Institution and Country
Schoen Klinik Berchtesgadener Land, Schoenau, Germany

Please state any competing interests or state 'None declared':
None declared

Please leave your comments for the authors below

I thank the authors for their comprehensive revision and advice to accept the manuscript for publication.

- Thank you for your valuable comments which have helped us improve the manuscript.

Reviewer: 3

Reviewer Name
Jana De Brandt

Institution and Country
UHasselt, Belgium

Please state any competing interests or state 'None declared':
None declared

Please leave your comments for the authors below

Dear authors,

Thank you for adapting the manuscript based on the reviewers suggestions. For me the manuscript is ready for publication after adapting the following comment. Thank you and good luck.

- *Many thanks for your comments, and for providing further clarification on this point.*

Regarding one of my comments:

Comment: In general I really like the fact that you take over the exact sayings of the participants. But for non-native English speakers/readers some of the expressions used by the participant might be difficult to understand. Is there any solution for this?

Response: We are glad that you like the verbatim use of quotes. We feel this is important to ensure the participant's voice is heard. We would be happy to put further clarifications regarding specific statements (as above), but its difficult for us to know exactly what needs further clarification without highlighting specific phrases.

Comment: I would address the following sentences in the same way as you did with the previous

sentence (using [...] for an explanation) to ensure all non-native English understand the sentences. I have put the more difficult wordings (in my opinion) after the arrow.

- *Many thanks. We have added clarification for these terms in to square brackets, and added a comment to the results section stating the purpose of these explanations.*

- 'I wouldn't call myself twinkle toes' (participant 3) ==> 'twinkle toes'
- 'I can't dance for toffee, and I can't sing, but hey (laughs) who cares.' (participant 1). ==> 'can't dance for toffee'
- 'I'm learning something new and it's also, you come in and learn a dance that six weeks ago you never thought you'd get your feet around.' ==> 'you'd get your feet around'
- '...before you know it, we're all in stiches. You know it's something, you know. Its promoted in the group. You can feel it. You get the buzz. You have a good laugh and, apart from doing your dancing you're having fun, even with the dancing you're having fun.' (participant 3) ==> 'we're all in stiches"
- 'we're like a small knit family' ==> 'knit family'